# Differences in the Volatile Profile of Apple Cider Fermented with *Schizosaccharomyces pombe* and *Schizosaccharomyces japonicus*

Nicola Ferremi Leali [1], Elisa Salvetti [1,2], Giovanni Luzzini [1], Andrea Salini [1], Davide Slaghenaufi [1], Salvatore Fusco [1], Maurizio Ugliano [1], Sandra Torriani [1,*] and Renato L. Binati [1]

[1] Department of Biotechnology, University of Verona, Strada Le Grazie 15, 37134 Verona, Italy; nicola.ferremileali@univr.it (N.F.L.); elisa.salvetti@univr.it (E.S.); giovanni.luzzini@univr.it (G.L.); andrea.salini@univr.it (A.S.); davide.slaghenaufi@univr.it (D.S.); salvatore.fusco@univr.it (S.F.); maurizio.ugliano@univr.it (M.U.); renato.lealbinati@univr.it (R.L.B.)

[2] VUCC-DBT, Verona University Culture Collection–Department of Biotechnology, University of Verona, Strada Le Grazie 15, 37134 Verona, Italy

[*] Correspondence: sandra.torriani@univr.it; Tel.: +39-045-8027921

**Abstract:** In this study, two strains of *Schizosaccharomyces pombe* (NCAIM Y01474[T] and SBPS) and two strains of *Schizosaccharomyces japonicus* (DBVPG 6274[T], M23B) were investigated for their capacity to ferment apple juice and influence the volatile compounds of cider compared to *Saccharomyces cerevisiae* EC1118. The ethanol tolerance and deacidification capacity of *Schizosaccharomyces* yeasts could make them potential substitutes for the commonly used *S. cerevisiae* starter cultures. Despite different time courses (10–30 d), all strains could complete the fermentation process, and *Schizosaccharomyces* strains reduced the concentration of malic acid in the apple juice. Results indicated that each yeast exerted a distinctive impact on the volatile profile of the apple cider, giving final products separated using a principal component analysis. The volatile composition of the cider exhibited significant differences in the concentration of alcohols, esters, and fatty acids. Particularly, the flocculant strain *S. japonicus* M23B increased the levels of ethyl acetate (315.44 ± 73.07 mg/L), isoamyl acetate (5.99 ± 0.13 mg/L), and isoamyl alcohol (24.77 ± 15.19 mg/L), while DBVPG 6274[T] incremented the levels of phenyl ethyl alcohol and methionol up to 6.19 ± 0.51 mg/L and 3.72 ± 0.71 mg/L, respectively. A large production of terpenes and ethyl esters (e.g., ethyl octanoate) was detected in the cider fermented by *S. cerevisiae* EC1118. This study demonstrates, for the first time, the possible application of *S. japonicus* in cider-making to provide products with distinctive aromatic notes".

**Keywords:** apple cider; *Schizosaccharomyces pombe*; *Schizosaccharomyces japonicus*; fermentation; malic acid; glycerol; volatile organic compounds

## 1. Introduction

Cider is an alcoholic beverage produced by the fermentation of apple juice. The greatest production of cider is in Europe; however, in recent years, its production has aroused an increased interest, both in Eastern European nations without a history of cider consumption and in the Northeast and Mid-Atlantic regions of the United States [1,2]. Becot and colleagues [3] reported that hard cider production has increased significantly in the United States, with an annualized growth rate of 50% between 2009 and 2014 with a total revenue of $292.5 million in 2014. The increased popularity of apple cider can be attributed to its freshness, sensory quality, and various other factors, including the territoriality, drinking occasion, and product information [2]. Furthermore, cider is rich in bioactive compounds, such as polyphenols, hydroxycinnamic acids, vitamin C, anthocyanins, and insoluble fibers [4]. The polyphenols have essential sensory characteristics, such as color, bitterness, astringency, and colloidal stability, while the fibers affect the viscosity of the row juice [1]. Additionally, cider is appropriate for consumers adopting a gluten-free vegetarian or vegan diet.

The sensory quality of cider is affected by several parameters, such as the apple cultivar, ripening stage, microbial strains, and processing conditions [5]. Alcoholic fermentation is the most crucial step in the cider-making process. The traditional spontaneous fermentation of apple juice is initially driven by non-*Saccharomyces* yeasts, such as *Hanseniaspora uvarum*, followed by *Saccharomyces* yeasts, originating from fruit and cider-making equipment. Yeasts convert fermentable carbohydrates into various end-products, including alcohol, carbon dioxide, and organic acids, reducing the dangerous growth of spoilage and pathogenic organisms. Further, secondary metabolites resulting from yeast activities, such as esters, higher alcohols, and phenolic compounds, provide novel and appealing flavors to cider [6,7].

To better control the fermentation process and have more predicted outcomes, the inoculation of selected starter cultures of *Saccharomyces cerevisiae* to carry out the alcoholic fermentation in apple juice is now one of the most common practices in cider-making. *Saccharomyces cerevisiae* was chosen specifically because it can quickly convert sugar to alcohol. However, the resulting cider has a simple aroma and a sour taste [8]. Therefore, the influence of non-*Saccharomyces* yeast species on apple cider's chemical composition and aroma complexity has also been investigated. Single or mixed culture fermentations of non-*Saccharomyces* yeasts (e.g., *Torulaspora delbrueckii*, *Zygosaccharomyces bailii*, *Lachancea thermotolerans*, *Starmerella bacillaris*, *Rhodotorula mucilaginosa*, and *Debaryomyces hansenii*) with *S. cerevisiae* have successfully been applied to produce apple cider with an improved sensory quality [8–11]. These studies highlighted that the fermentation by unconventional yeasts mainly affects the volatile composition of apple cider, producing different amounts of alcohols, esters, and fatty acids, thus making distinctive and unique products.

Among the non-*Saccharomyces* yeasts, the fission yeast *Schizosaccharomyces pombe* has been proposed as a potential substitute for *S. cerevisiae* in the production of apple cider (with lower alcohol content) and apple wine (higher alcoholic content > 8.5%) due to its similar fermentation ability [12,13]. This yeast has been used in wine production for a long time, mainly for its distinctive deacidifying activity, making very acidic wines smoother [14]. Indeed, it shows a rapid malic acid deacidification by converting malic acid into $CO_2$ and pyruvic acid, which is further converted to ethanol via the malo-ethanolic pathway. *S. pombe* has also been a useful tool to reduce the malic acid content of cider, resulting in a lowered sourness compared to cider produced with *S. cerevisiae* [14]. In addition to biological deacidification, *S. pombe* is gaining increasing attention in wine and other fermented beverage industries due to its ability to influence fermentation quality parameters and solve specific challenges (e.g., sourness and vinyl phenol production) [13,14]. Selected strains of this species produce valuable metabolites in cider, such as glycerol, esters, and polysaccharides, which are important determinants of sensory perception. Even though *S. pombe* is the most common species of the genus used in fermentative processes, *Schizosaccharomyces japonicus* is a promising species. This species has many properties like those of *S. pombe* (e.g., malic acid deacidification); however, to date, yeasts belonging to *S. japonicus* have not been utilized for cider-making.

To fill the knowledge gap on the fermentative capacity of *Schizosaccharomyces* fission yeasts, this study evaluated the suitability of two strains of *S. pombe* and two strains of *S. japonicus* for fermenting apple juice. Therefore, the aims were to investigate their fermentation performances and quantify the volatile and non-volatile compounds in the final cider compared with *S. cerevisiae* EC1118. The current research can provide insights into applying new microbial resources to produce apple cider with distinctive characteristics.

## 2. Materials and Methods

### 2.1. Yeast Strains and Inoculum Preparation

The yeast strains used in this study are listed in Table 1 (strain designations, used from here on for clarity, are also reported). *S. japonicus* Szj is a natural flocculant strain. All strains were maintained under cryo-preservation at $-80\,^{\circ}$C in 25% *v/v* glycerol in the Verona University Culture Collection–Department of Biotechnology (VUCC-DBT). They

were routinely grown at 27 °C in YPD broth (10 g/L yeast extract, 20 g/L bacteriological peptone, and 20 g/L dextrose).

**Table 1.** Strain designation, origin, and isolation source of the yeast strains used in this study.

| Species | Strain | Strain Designation | Origin and Isolation Source |
|---|---|---|---|
| *Schizosaccharomyces pombe* | NCAIM Y01474[T] | SzpT | NCAIM culture collection (Budapest, Hungary), isolated from Arak (a fermented beverage) macerate |
| *Schizosaccharomyces pombe* | SPBS | Szp | Agricultural and Food Sciences Department of Bologna University culture collection (Bologna, Italy), isolated from fruit syrup |
| *Schizosaccharomyces japonicus* | DBVPG 6274[T] | SzjT | DBVPG culture collection (Perugia, Italy), isolated from strawberry wine |
| *Schizosaccharomyces japonicus* | M23B | Szj | VUCC-DBT culture collection (Verona, Italy), isolated from grape must (Rovereto, TN, Italy) |
| *Saccharomyces cerevisiae* | EC1118 | Sc | Lallemand Inc. (Montreal, QC, Canada), isolated from French sparkling wine |

The inocula for the fermentation trials were prepared following the procedures of [15]. The yeasts were grown overnight in YPD broth at 27 °C, with shaking, to reach the early stationary phase. Next, cells were harvested by centrifugation at $3000\times g$ for 5 min, washed twice with physiological solution (0.9% $w/v$ NaCl), re-suspended in heat-treated commercial apple juice (Weissenhof, Vilpiano, Italy), and counted under a microscope using a Burker counting chamber to prepare the inoculum at a concentration of about $1 \times 10^6$ cells/mL. All reagents were purchased from Sigma-Aldrich (Milan, Italy).

## 2.2. Cider Fermentation Kinetics and pH Monitoring

The standardized inoculum of each yeast, prepared as described above, was transferred to a sterile 500-mL glass bottle containing 400 mL of apple juice (Weissenhof, Vilpiano, Italy). Fermentations were performed in triplicate, and bottles were incubated at 27 °C.

Fermentation kinetics were monitored by the daily weighing of the bottles, reflecting the mass loss due to $CO_2$ release. Fermentations were stopped when the daily weight loss was less than 0.05 g. pH measurements were performed at the beginning, middle, and end of fermentations using the pH-meter Crison Basic 20 (Hach-Lange, Barcelona, Spain).

## 2.3. High-Performance Liquid Chromatography (HPLC) Analysis

An HPLC analysis was carried out at the beginning and at the end of fermentations to quantify the sugars (glucose, fructose, and sucrose), organic acids (malate, acetate, and citrate), polyols (glycerol and sorbitol), and ethanol using the Extrema LC-4000 system (Jasco, Cremella, Italy) coupled with a refractive index detector RI-4030 (Jasco) set to 35 °C. Analytes were separated using a RezexTM ROA-Organic Acid H + (8%) column (300 × 7.8 mm; Phenomenex, Castel Maggiore, Italy) maintained at 80 °C and under an isocratic mobile phase (5 mM $H_2SO_4$) (Honeywell, Rodano, Italy) at a flow rate of 0.8 mL/min. The column had a SecurityGuardTM Cartridges, Carbo-H (4 × 3.00 mm) (Phenomenex) guard column. Before analysis, the samples underwent centrifugation at $6000\times g$ for 5 min, followed by filtration using 0.22 µm syringe filters (SPHEROS, LLG Labware, Meckenheim, Germany). Subsequently, the filtered samples were suitably diluted with 5 mM of $H_2SO_4$. The analytes were quantified using calibration curves and prepared within a 0.1 to 7 g/L concentration range.

### 2.4. Volatile Organic Compound Analysis

The procedure described by [16] was followed for quantifying alcohols, esters, and fatty acids, performing a solid-phase extraction (SPE). Fifty mL samples of apple cider with the addition of a 100 μL internal standard (2-octanol, 4.2 mg/L in ethanol) were diluted with 50 mL of deionized water. Before analysis, the SPE cartridge, BOND ELUT-ENV (Agilent Technologies, Santa Clara, CA, USA) was activated by eluting 20 mL of dichloromethane, followed by 20 mL of methanol, and equilibrated with 20 mL of water.

Then, the diluted cider was loaded by percolating it through the SPE cartridge. Sugar and polar compounds were eliminated by washing with 15 mL of water. Subsequently, volatile compounds were eluted with 10 mL of dichloromethane. The organic phase was then concentrated to 200 μL under a gentle nitrogen stream. The sample was then ready for GC injection.

A gas chromatography-mass spectrometry (GC-MS) analysis was carried out on an HP 7890A gas chromatograph coupled to a 5977B single quadrupole mass spectrometer (Agilent Technologies, Cernusco sul Naviglio, Italy). Two μL of the organic extract were injected in splitless mode at 250 °C. Separation was performed using a DB-WAX UI capillary column (30 m × 0.25 mm, 0.25 μm film thickness, Agilent Technologies) and helium (6.0 grade) as the carrier gas at 1.2 mL/min of constant flow rate, the oven was initially set at 40 °C for 3 min, then increased to 230 °C at 4 °C/min. The MS operated in electronic impact ionization mode (EI) set at 70 eV. The transfer line, ion source, and quadrupole temperature were set at 200, 230, and 150 °C, respectively. The acquisition mode was synchronous SCAN ($m/z$ 40–200) and single ion monitoring (SIM). Samples were analyzed in random order.

Free terpenes, norisoprenoids, and methyl salicylate were quantified using a solid-phase micro extraction (SPME), following the procedure of [17]. An aliquot of the sample (5 mL) was added to a 20-mL vial with 3 g of NaCl, 5 mL of deionized water, and 5 μL of internal standard 2-octanol (4.2 mg/L in ethanol). Sampling and injection were performed using a Gerstel MPS3 auto-sampler (Gerstel, Müllheim a der Ruhr, Germany), a 50/30 μm DVB/CAR/PDMS (divinylbenzene–carboxy–polydimethylsiloxane) fiber (Supelco, Bellefonte, PA, USA) was placed in the headspace of the vial sample for 60 min at 40 °C. The injection was performed by exposing the fiber to the GC inlet at 230 °C for 3 min. Chromatographic separation and MS acquisition were performed as described above for the major volatile compound analysis. In both methods, the quantification was done via seven-point calibration curves in the matrix.

### 2.5. Statistical Analysis

Data of analytical determinations on the GC-MS were compared via a one-way analysis of variance (ANOVA), followed by the post hoc Tukey's HSD (honestly significant difference) test, with the statistical significance threshold set at 95% ($p$-value < 0.05), and scaled before principal component analysis (PCA). GC-MS data were averaged, centered, and scaled by compound and hierarchically clustered by Ward's minimum variance method and the Euclidean distance metric by the ggplot2 function in R [18].

## 3. Results

### 3.1. Fermentations Kinetics

The course of the fermentations, represented by $CO_2$ release, is shown in Figure 1.

Fermentations of the *Schizosaccharomyces* strains showed a strain-dependent time course. For SzjT, Szp, and SzpT, the $CO_2$ release fell below 0.05 g in 23 days. The fermentation rate of Szj was slower and took 30 days to stabilize. Nevertheless, the control fermentation with Sc showed a significantly faster nine-day fermentation.

The initial pH of the apple juice was 3.70 and decreased throughout the fermentations for the *S. pombe* strains until 3.27 ± 0.05 (Table 2). On the contrary, SzjT increased the pH slightly, and the strain Szj increased the pH to 4.42 ± 0.21 after 30 days.

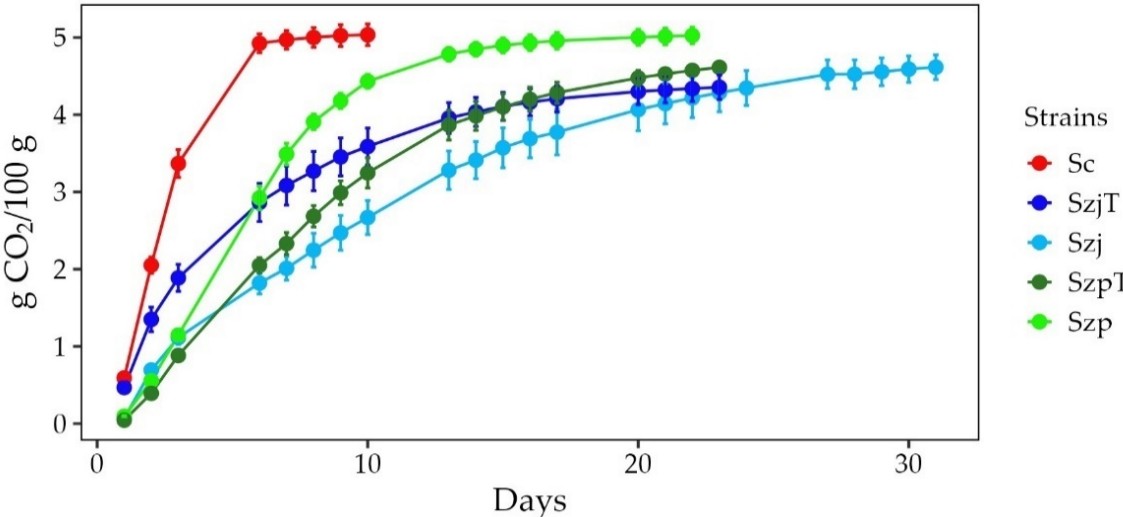

**Figure 1.** Fermentation kinetics of *Schizosaccharomyces pombe* SzpT and Szp, *Schizosaccharomyces japonicus* SzjT and Szj, and *Saccharomyces cerevisiae* Sc in apple juice. Data are the averages of three independent fermentations.

**Table 2.** Chemical composition of apple cider fermented with *Schizosaccharomyces pombe* SzpT and Szp, *Schizosaccharomyces japonicus* SzjT and Szj, and *Saccharomyces cerevisiae* Sc. Data are the averages of three independent fermentations. Different letters for the same data series indicate a significant difference in Tukey's HSD test ($p < 0.05$). N.D. = not detected.

| Compound (g/L) | Apple Juice | Strain | | | | |
| --- | --- | --- | --- | --- | --- | --- |
| | | SzpT | Szp | SzjT | Szj | Sc |
| Fructose | 91.14 | N.D. | N.D. | N.D. | N.D. | N.D. |
| Glucose | 40.42 | N.D. | N.D. | N.D. | 0.17 ± 0.09 | N.D. |
| Sucrose | 13.96 | N.D. | N.D. | N.D. | N.D. | N.D. |
| Citrate | N.D. | 0.34 ± 0.01 [a] | 0.37 ± 0.03 [a] | N.D. | N.D. | N.D. |
| Malate | 3.25 | 0.80 ± 0.01 [b] | 0.79 ± 0.02 [b] | 0.69 ± 0.01 [c] | 0.11 ± 0.02 [d] | 3.22 ± 0.02 [a] |
| Glycerol | N.D. | 3.89 ± 0.18 [c] | 4.01 ± 0.10 [c] | 7.21 ± 0.53 [b] | 8.46 ± 0.57 [a] | 3.10 ± 0.27 [c] |
| Sorbitol | N.D. | 5.89 ± 1.66 [a] | 5.57 ± 1.50 [a] | 4.39 ± 1.03 [a] | 6.56 ± 0.65 [a] | 6.41 ± 1.20 [a] |
| Acetate | N.D. | 0.28 ± 0.02 | N.D. | N.D. | N.D. | N.D. |
| Ethanol (v/v, %) | N.D. | 6.93 ± 0.35 [ab] | 7.16 ± 0.24 [ab] | 6.07 ± 0.32 [c] | 6.53 ± 0.11 [bc] | 7.27 ± 0.27 [a] |
| pH | 3.70 | 3.27 ± 0.05 [c] | 3.54 ± 0.13 [bc] | 3.73 ± 0.04 [b] | 4.42 ± 0.21 [a] | 3.67 ± 0.01 [b] |

*3.2. Fermentation Parameters*

The content of the residual sugars (fructose, glucose, and sucrose), organic acids (citrate, acetate, and malate), polyols (glycerol and sorbitol), and ethanol were quantified in apple juice and cider using HPLC. The results of the chemical analysis are summarized in Table 2 and Supplementary Table S1.

All strains consumed completely the fructose, glucose, and sucrose in the apple juice except Szj, which left a residual trace of glucose.

The ethanol concentration was similar in cider fermented with the *S. pombe* strains and comparable to the control Sc (7.27 ± 0.28% *v/v*), while it was slightly reduced in cider with the *S. japonicus* strains.

As expected, all *Schizosaccharomyces* strains metabolized malate, and Szj was the most efficient consumer, with only 0.11 ± 0.02 g/L of residue in the cider after 30 days of fermentation. Instead, *S. cerevisiae* did not affect the malate concentration.

Remarkably, the final glycerol concentration was significantly higher in the fermentation with *S. japonicus* and, among the two strains, Szj produced the most significant level

(8.46 ± 0.57 g/L). Regarding sorbitol, it ranged from 4.39 ± 1.03 g/L to 6.56 ± 0.65 g/L for SzjT and Szj, respectively, without significant differences among strains.

### 3.3. Volatile Organic Compounds in Apple Cider

The evaluation of apple cider through GC-MS allowed for the detection of 29 volatile organic compounds (VOCs) above the limit of quantification (Supplementary Table S2). They belong to the following six chemical families: esters (eight compounds), alcohols (five), terpenes (six), fatty acids (three), benzenoids (five), sulfur-containing compounds (one), and norisoprenoids (one).

The overall content of compounds in each family showed that *S. japonicus*, particularly the strain Szj, exhibited the most significant ester and alcohol concentrations. The *S. pombe* strains produced the lowest concentration of esters and fatty acids, while Sc increased the levels of fatty acids.

To better visualize each yeast strain's influence on the apple cider's VOC profile, the heat plot in Figure 2 shows the relative abundance of the single compounds and the strain clusters according to their similarity.

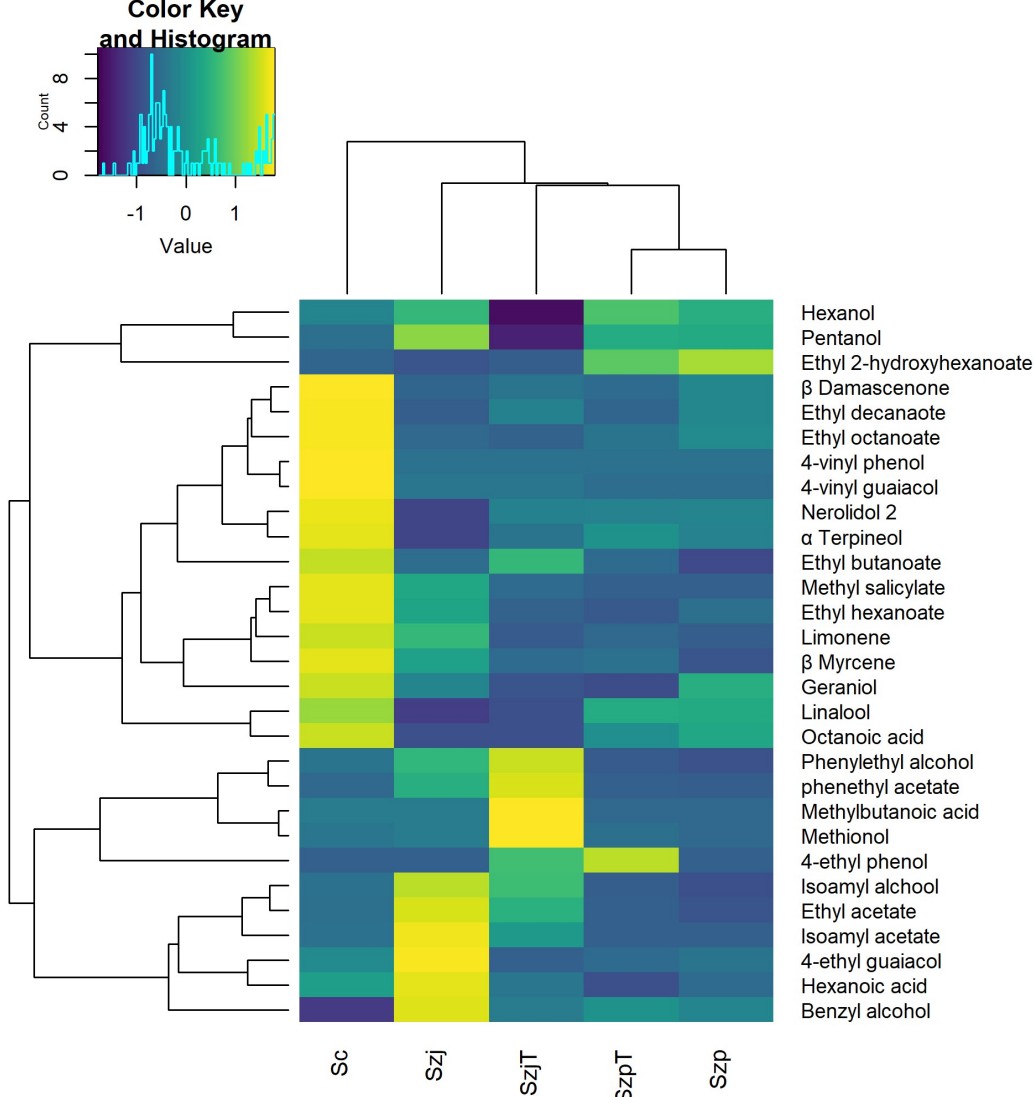

**Figure 2.** Heat plot and hierarchical cluster analysis of the relative abundance of volatile organic compounds in apple cider fermented with *Schizosaccharomyces pombe* SzpT and Szp, *Schizosaccharomyces japonicus* SzjT and Szj, and *Saccharomyces cerevisiae* Sc. The Color Key Histogram identifies the values and their occurrence in the heatmap. The quantitative data used are available in Supplementary Table S2.

The hierarchical cluster analysis formed three clusters that separated the three yeast species. Sc increased the concentration of most of the molecules, especially the terpenes and fatty acids, while *S. japonicus* influenced a few essential compounds. An intraspecific diversity was observed for the *Schizosaccharomyces* strains. Szj caused a remarkable increase in ethyl acetate, isoamyl acetate, isoamyl alcohol, and benzyl alcohol, while SzjT strongly accentuated the levels of phenethyl alcohol, methionol, phenethyl acetate, and 3-methyl-butanoic acid. Regarding *S. pombe*, both strains produced ethyl 2-hydroxy hexanoate, while SzpT also produced the highest quantity of hexanol.

The 29 VOCs were subjected to a principal component analysis (PCA) to visualize the characteristics of the cider obtained with the tested yeast strains (Figure 3). The first and second components represented in the graph accounted for 72.3% of the total variation (PC1 = 44.6% and PC2 = 27.7%). A good reproducibility of the replicates was observed as they were positioned close to each other.

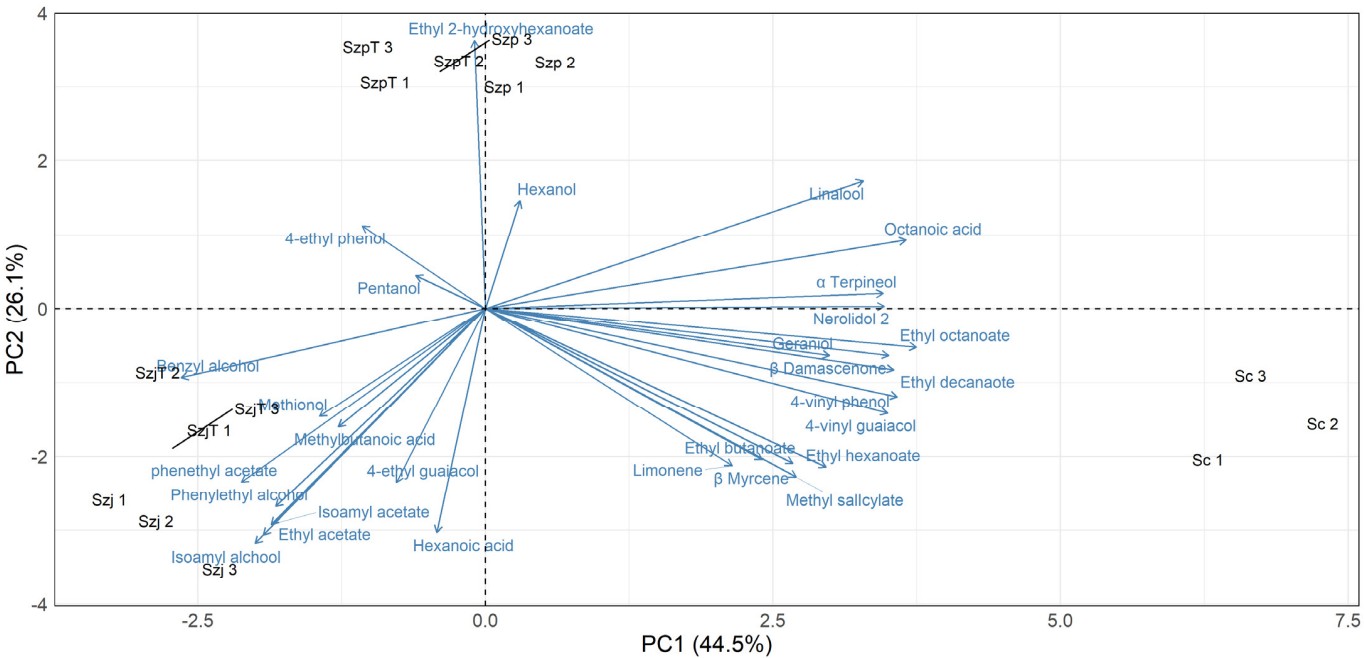

**Figure 3.** Principal component analysis biplot derived from volatile organic compounds of apple cider fermented with *Schizosaccharomyces pombe* SzpT and Szp, *Schizosaccharomyces japonicus* SzjT and Szj, and *Saccharomyces cerevisiae* Sc. The three replicates are indicated by the strain code followed by a Roman number.

The main differences highlighted in the heat plot were also clearly acknowledged in the PCA. The cider fermented with the three species was differentiated: the first component (PC1) discriminated the cider fermented by Sc (located in the lower right quadrant) from that fermented by *S. japonicus* strains (lower left quadrant), mainly for the different content in terpenes and esters; the second component (PC2) separated *S. pombe* strains (upper quadrants) from the strains of the other species, particularly for the hexanol, pentanol, and ethyl 2-hydroxyhexanoate.

## 4. Discussion

Apple cider is a fermented beverage resulting from the activity of yeasts in apple juice; thus, the choice of starter cultures for its production is fundamental [5]. Besides *S. cerevisiae*, non-*Saccharomyces* yeasts, including *S. pombe*, have stimulated interest in the beverage industry as an innovative way of creating new flavors. However, the use of the *Schizosaccharomyces* fission yeasts is so far yet to be well investigated. Therefore, this study shows the effect of the *S. japonicus* strains SzjT and Szj and *S. pombe* strains SzpT and Szp

on the apple cider's chemical and volatile composition, evaluated in comparison with that of *S. cerevisiae* Sc.

The fermentation kinetics of the strains SzjT, SzpT, and Szp shared a similar trend, completing the fermentation after 23 days. Analogous results were reported for *S. pombe* 3796 (18–20 days) [14], while the fermentation time course of *S. cerevisiae* V1116 was significantly slower (16 days) compared to that of the *S. cerevisiae* Sc analyzed in this study (10 days). The slower fermentation kinetics of *S. japonicus* Szj (30 days) might be correlated with the robust flocculation of this strain, which limited the contact between cells and substrate [19]. Fermentation time influences aromatic compounds' production, determining the ratio of higher alcohols to esters; a longer fermentation increases the content of fatty acids, diminishing the amount of soluble solids [20].

The strain Szj stood out for the pH increase (0.71) of cider, consistent with the almost total (96.62%) utilization of malic acid, the major organic acid in apples. Such a high pH could pose safety questions. Nevertheless, pH is often adjusted during cider production to below 3.8 to avoid microbiological spoilage [21]. However, unlike the data obtained by [14], the *S. pombe* strains did not significantly modify the pH, even if they reduced malic acid to about 76%. pH is a critical factor in fermentation as it influences the final product, such as its flavor, color, and aroma. As fermentation progresses, the yeast gradually converts sugars into alcohol and other metabolites, and the accumulation of organic acids is also reduced. This results in a gradual increase in the pH of the fermentation solution. Increased pH values can lessen the astringent puckering, acidity, and harsh apple flavor [14].

All strains depleted glucose, fructose, and sucrose confirming their capacity to metabolize these sugars, with a preference for glucose, as highlighted by the half-fermentation analysis data (Supplementary Table S1), where the fructose–glucose ratio increased during fermentation from 2.25 in apple juice to 3.04–7.49 in cider, as previously observed by He and colleagues [14]. The apple juice concentration of available sugars is directly proportional to the theoretical ethanol yield, since all strains consumed the available sugars, the ethanol concentration was similar after fermentation (approx. 6.78% *v/v*) and matched the previous study by He and colleagues [14]. Therefore, following the cider definition provided by the European Cider and Fruit Wine Association (AIVC, aicv.org), who define ciders "as alcoholic beverages, with a % *v/v*, of between 1.2% and 8.5% *v/v*" [22], the fermented juice obtained in the present study can be defined as apple cider.

As regards polyols, glycerol was produced by all tested strains, and the *S. japonicus* strains were the greatest producers. This capacity has been well established in wine, where *S. japonicus* produced a quantity of glycerol, which was 2-fold higher than those released by *S. cerevisiae* [23]. Interestingly, the cider fermented by Szj showed the highest pH and the utmost glycerol concentration, probably because of the increased activity of aldehyde at a higher pH value. Indeed, this step generates NADH, which is re-oxidized through the glycerol pathway in anaerobiosis conditions [24]. The present investigation showed no significant difference between Sc and *S. pombe* strains (3.10 ± 0.27–4.01 ± 0.10 g/L). The amount of ethanol and glycerol in wine favorably affects the wine's fullness (or "body") on the palate. Since cider typically has a lower ethanol content (8–8.5%) than wine, significant glycerol formation is important. Glycerol positively impacts the sensory properties of ciders, assuring their fullness of taste and smoothness [5]. Also, sorbitol was produced by all strains in statistically equivalent amounts (4.41–6.56 g/L); Duvnjak and colleagues [25] reported that this polyol is produced by *S. cerevisiae* in the presence of fructose after the consumption of glucose, as observed here in the half-fermentation analysis data (Supplementary Table S1). Sorbitol can be used as a substitute for sugars, providing a sweet taste without increasing the calorie count and beneficially affecting cider taste [26].

The investigation of VOCs is crucial, as they contribute significantly to the final sensory quality of fermented beverages. The most common ones identified in apple cider belong to different chemical families, including esters, higher alcohols, fatty acids, and, in smaller quantities, phenolic compounds and terpenes [20]. Their concentration varies depending on

the apple cultivar and level of ripeness, but these volatile compounds are mainly produced through yeast alcoholic fermentation as secondary metabolites [9]. An analysis of VOC profiles revealed significant variations of relevant aroma-active molecules, such as esters, alcohols, and fatty acids, depending on the yeast. *S. japonicus* Szj was the greatest producer of esters; in particular, the ethyl acetate and isoamyl acetate concentration was 2-fold greater than that in *S. japonicus* SzjT. This might be correlated with the prolonged fermentation time, which leads to increased concentrations of these compounds in cider [20] and the expression levels of yeast alcohol acetyltransferase genes [27]. Excessive ethyl acetate levels should, however, be avoided as this compound can impart nail polish aromas that are detrimental to the overall quality at high concentrations [28]. Interestingly, SzjT increased the phenylethyl alcohol with a floral and fresh aroma. Sc produced more ethyl acetate than *S. pombe* strains, as reported previously in apple cider [14]. During fermentation, ethanol and acetyl-CoA are converted to ethyl acetate, one of the most important esters influencing apple wine flavor. Its aroma is particularly noticeable in younger ciders and adds to the overall impression of fruitiness [29]. Cider fermented by *S. cerevisiae* showed the greatest concentration of ethyl butanoate, hexanoate, octanoate, and decanoate. Ethyl hexanoate and octanoate are essential cider compounds and ethyl acetate [30,31]. However, an ethyl acetate concentration over 200 mg/L may negatively affect quality, hiding varietal aromas and simplifying the aroma composition [32].

The higher alcohol concentration is greater in cider fermented by the *S. japonicus* strains. Szj produced three times more isoamyl alcohol, while SzjT accumulated four times more methionol compared to the reference strain Sc. Isoamyl alcohol provides banana aromatic notes and can be generated from amino acids in yeast-based bioconversion processes [33]. Yeasts produce alcohols as a by-product of the nitrogen assimilation process, where free amino acids are absorbed through the Ehrlich pathway. This pathway is mainly studied in *S. cerevisiae*; however, the same reaction steps are shared with non-*Saccharomyces* yeasts [34]. Depending on the aromatic context, high levels of alcohols in apple cider may suppress fruity and woodsy notes, thus harming the aroma. Alcohols also cause a strong taste and odor when present in concentrations greater than 300 mg/L [35].

Fatty acids can play a role in the aroma of apple cider, enhancing the complexity and aromatic balance since they are characterized by notes of fruit, cheese, and rancidity, even if they typically have high odor detection thresholds [28]. In this study, three fatty acids (hexanoic acid, octanoic acid, and 3-methylbutanoic acid) were quantified in the apple cider. The *S. japonicus* strain Szj produced the highest amount of hexanoic acid, while Sc produced up to $25 \pm 0.8$ mg/L of octanoic acid.

Unpleasant, medicinal, phenolic aromatic notes characterize vinyl phenols. Yeasts produce 4-vinyl phenol (4VP) and 4-vinyl guaiacol (4VG) via the decarboxylation of hydroxycinnamic acids. Since apple juice is rich in precursor cinnamic acids, in cider the concentrations of 4VP and 4VG can reach high levels, considerably impacting the organoleptic quality [36]. Indeed, data demonstrated rather high concentrations of 4VP and 4VG for *S. cerevisiae*, exceeding the odor threshold. Instead, both *Schizosaccharomyces* species produced at least 10-times-lower quantities. Furthermore, there was a statistically significant difference between *S. japonicus* and *S. pombe* with the latter having produced less than 5 µg/L of both 4VP and 4VG. The use of *Schizosaccharomyces*, therefore, seems potentially interesting in limiting the production of the off-flavors 4VG and 4VP.

As regards the terpenes, these compounds are associated with the flowery flavor of cider and their relevance is linked to their low odor threshold [37]. The cider fermented with *S. cerevisiae* showed the highest concentration of terpenes, resulting from bioconversion and glycoside hydrolysis [20]. The terpene content of the cider produced with *S. pombe* was reported to vary with yeast strains and apple varieties [14]. Our data showed that *S. pombe* strains produced higher levels of terpenes compared to *S. japonicus.* This could lead to products more characterized by citrus-floral odor notes. However, the terpene levels found in samples were below the odor threshold, suggesting that they were not contributing directly to the cider aroma. However, it is possible that terpenes at subthresh-

old concentrations contribute to olfactory perception through additive and synergistic effects [38].

Regarding the sensory role of the volatile compounds, we observed that 14 metabolites were found in such concentrations to be aromatically active (Table 3). Among these, 11 compounds were of fermentative origin (esters, alcohols, and fatty acids), the megastigmane norisoprenoid β-Damascenone and the two vinyl phenols 4VG and 4VP. None of the terpenes, including linalool, found in higher concentrations up to 5.67 μg/L, exceeded the odor threshold.

**Table 3.** Odor activity value (OAV †) of odor-active compounds of apple cider fermented with *Schizosaccharomyces pombe* SzpT and Szp, *Schizosaccharomyces japonicus* SzjT, and Szj, and *Saccharomyces cerevisiae* Sc.

| Compounds | OT ‡ | Strain | | | | |
|---|---|---|---|---|---|---|
| | | **SzpT** | **Szp** | **Szjt** | **Szj** | **Sc** |
| *Esters* | | | | | | |
| Ethyl acetate | 12 mg/L | 2.71 | 1.47 | 15.08 | 26.29 | 5.27 |
| Ethyl butanoate | 20 μg/L | 6.27 | 5.22 | 9.14 | 6.39 | 11.29 |
| Ethyl hexanoate | 14 μg/L | 9.13 | 12.26 | 10.63 | 19.77 | 32.05 |
| Ethyl decanoate | 200 μg/L | 0.25 | 0.56 | 0.51 | 0.17 | 1.76 |
| Isoamyl acetate | 30 μg/L | 6.12 | 4.83 | 72.23 | 199.8 | 25.81 |
| 2-phenethyl acetate | 108 μg/L | 1.29 | 1.21 | 10.87 | 6.1 | 1.82 |
| *Alcohols* | | | | | | |
| Isoamyl alcohol | 30 mg/L | 1.42 | 0.83 | 5.92 | 8.26 | 2.37 |
| Phenylethyl alcohol | 390 μg/L | 7.38 | 6.99 | 15.89 | 12.42 | 8.8 |
| *Fatty acids* | | | | | | |
| Hexanoic acid | 420 μg/L | 4.29 | 6.48 | 7.39 | 19.14 | 10.74 |
| Octanoic acid | 500 μg/L | 21.61 | 28.07 | 4.08 | 4.29 | 50.1 |
| 3-Methylbutanoic acid | 250 μg/L | 0.07 | 0.14 | 9.45 | 1.2 | 1.22 |
| *Norisoprenoids* | | | | | | |
| β-Damascenone | 0.05 μg/L | 14.4 | 20.8 | 16.33 | 13.1 | 49.73 |
| *Benzenoids* | | | | | | |
| 4-vinyl guaiacol | 40 μg/L | 0.08 | 0.1 | 0.29 | 0.33 | 3.76 |
| 4-vinyl phenol | 180 μg/L | 0.02 | 0.02 | 0.03 | 0.03 | 2.09 |

† OAV has been calculated as the concentration divided by the odor threshold. ‡ OT is short for odor threshold. OT values are referred to Bingman and colleagues [39].

## 5. Conclusions

Like other fermented beverages, the sensory quality of apple cider is strictly dependent on the yeast used for the alcoholic fermentation. The results of this study indicate that the tested *Schizosaccharomyces* yeasts strongly affected the chemical and volatile composition of apple cider in a strain-specific manner, providing final products with distinctive characteristics concerning *S. cerevisiae*, used as a reference. All the *S. pombe* and *S. japonicus* strains completed the fermentation process, despite different time courses, and reduced the malic acid concentration, leading to a sourness decrease in the cider. The final products had an ethanol content of approx. 6.78% *v/v*, which is thus in the range defined by the AICV [24]. The VOC profile of the cider fermented with the flocculant *S. japonicus* strain was very different from those of the other tested strains, showing greater concentrations of esters, such as ethyl acetate and isoamyl acetate, and higher alcohols, such as isoamyl alcohol. Therefore, further studies are needed to assess the impact of *Schizosaccharomyces* strains on cider sensory properties, defining the consumer acceptance and preference. Moreover, the fermentation of sequentially or co-inoculated mixed strains might improve the cider's volatile composition. This study demonstrates, for the first time, the possible application of *S. japonicus* in cider-making to provide products with distinctive aromatic notes.

**Supplementary Materials:** The following supporting information can be downloaded at: https://www.mdpi.com/article/10.3390/fermentation10030128/s1, Table S1: Chemical composition of apple cider fermented with *Schizosaccharomyces pombe* SzpT and Szp, *Schizosaccharomyces japonicus* SzjT and Szj, and *Saccharomyces cerevisiae* Sc at half fermentation. Data are the averages of three independent fermentations. Different letters for the same data series indicate a significant difference in Tukey's HSD test ($p < 0.05$). N.D. = not detected; Table S2: Volatile organic compounds ($\mu$g/L) in apple cider fermented with *Schizosaccharomyces pombe* SzpT and Szp, *Schizosaccharomyces japonicus* SzjT and Szj, and *Saccharomyces cerevisiae* Sc.

**Author Contributions:** Conceptualization, R.L.B., E.S. and S.T.; methodology, N.F.L., A.S., D.S., G.L. and R.L.B.; formal Analysis, R.L.B., E.S. and M.U.; investigation, N.F.L., G.L., A.S., D.S. and R.L.B.; resources, S.T., M.U. and S.F.; writing—original draft preparation, N.F.L., E.S. and R.L.B.; writing—review and editing, N.F.L., E.S., D.S., M.U., S.F. and R.L.B.; visualization and supervision, S.T. and E.S.; funding acquisition, S.T. All authors have read and agreed to the published version of the manuscript.

**Funding:** This work has been partly developed within the research program "Dipartimento di Eccellenza 2023–2027" (Italian Ministry of University and Research). The PhD scholarship of N.F.L. was funded by REACT-EU FSE in the frame of PON "Ricerca e Innovazione" 2014–2020 (DM 1061/2021). Codice BIO13, DOT1340225, Borsa 1 CUP B39J21026610001.

**Data Availability Statement:** The data presented in this study are available on request from the corresponding author.

**Acknowledgments:** The authors would like to thank Michele Giacomi for technical assistance and Fausto Gardini for providing the *Schizosaccharomyces pombe* strain SPBS.

**Conflicts of Interest:** The authors declare no conflicts of interest. The funders had no role in the design of the study, in the collection, analyses, or interpretation of data, in the writing of the manuscript, or in the decision to publish the results.

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
