# Peer review of "Differences in the Volatile Profile of Apple Cider Fermented with Schizosaccharomyces pombe and Schizosaccharomyces japonicus"

_fermentation, doi:10.3390/fermentation10030128_

Round 1

Reviewer 1 Report

Comments and Suggestions for Authors

Manuscript ID fermentation-2876027

The authors compared two Schizosaccharomyces japonicus (DBVPG 6274T, M23B) and two Schizosaccharomyces pombe (NCAIM Y01474T and SBPS) strains with EC1118 in apple juice fermentations and analyzed the resulting volatile compounds of cider.

The authors argue that Schizosaccharomyces strains could serve as starter cultures in cider making.

>>> This is a bold statement given the characteristic off-flavor Schizosaccharomyces strains produce – namely 4 VG and 4 EG.

The authors state that all strains were able to complete the fermentation process, despite different time courses, and reduce the concentration of malic acid in apple juice.

>>> here it should be noted that EC1118 has no real capacity to metabolize malic acid.

The authors state that “ Sz. japonicus M23B increased the levels of ethyl acetate”

>>> this appears to be a positive statement. Here the amounts need to be stated as too much of it is clearly an off-flavor.

The authors conclude “ This study 23 demonstrates, for the first time, the possible application of Sz. japonicus in cider-making to provide 24 products with distinctive aromatic notes.“

>>> distinctive as in distinctively unpleasant?

Comments:

Intro:

“including perceived health effects“

>>> it should be clear that there are no health benefits with alcoholic beverages. Even alluding to perceived benefits is not sound.

Materials:

L84 the inocula

L88: “and counted under a microscope” … really?

L89: “6 Log cells/mL.“ State that 1x 106 cells/mL were meant.

L94: “Fermentations were performed in triplicate, and bottles were incubated at 27 °C.
>>> 27 °C – really? That is far higher than other cider fermentations. Any reason for that? This has a severe influence on the fermentation outcomes. Also, EC1118 is not best at 27°C.

L96: reflecting the loss; add “of mass”

2.3. HPLC

Sucrose and acetate were not mentioned. Were they not analyzed?

Results

Ok there was 23-30 day fermentations at 27°C and fermentations were barely finished. This really needs to be discussed as this is a downside of using these strains – and it needs to be repeated at lower temperatures.

pH: it was noted that all strains reduced malic acid content: then why was there no pH increase in S. pombe fermentations?

A real drawback of the analysis is the presence of maltose: are you really sure?

Such an amount of maltose is an indication that the juice was not solely apple juice. What really strikes me is the negligence to analyze sucrose as this is a major sugar in apple juice. Without such an analysis the paper is not publishable. Again, if there was no sucrose in the juice it was not apple juice.

The paper is not data rich. Thus I suggest to move all supplemental material to the main text for better visibility.

The authors did not comment on the nitrogen content of the juice nor if they added ammonium or other add-ons to increase YAN.

Discussion

“The investigation of VOCs is crucial, as they contribute significantly to the final sen-290 sory quality of fermented beverages“

Agreed so please analyse the off-flavors and acetate.

Comments on the Quality of English Language

none

Reviewer 2 Report

Comments and Suggestions for Authors

The manuscript investigates the fermentative capacity and impact on volatile compounds of two strains of Schizosaccharomyces japonicus (DBVPG 6274T, M23B) and two strains of Schizosaccharomyces pombe (NCAIM Y01474T and SBPS) in comparison to the commonly used Saccharomyces cerevisiae EC1118 during apple juice fermentation. The key findings suggest that Schizosaccharomyces yeasts, due to their ethanol tolerance and deacidification capacity, have the potential to be substitutes for S. cerevisiae in cider production. The paper can be considered again after the revisions requested, as detailed below.

While the introduction is well-constructed, there are a few areas that could be enhanced for further clarity and depth:

#1 Provide a brief overview or context on Schizosaccharomyces yeasts for readers who might not be familiar with them. This could include general characteristics, metabolic pathways, or their role in other fermentation processes.

#2 Connect the information on bioactive compounds in cider (polyphenols, hydroxycinnamic acids, etc.) more explicitly to the overall quality and sensory characteristics of cider. Help the reader understand why these compounds are important in the context of the study.

#3 After discussing the factors influencing cider quality, make a smoother transition to the central focus of the study: the role of Schizosaccharomyces yeasts in cider fermentation. This could involve a clearer statement or bridge between general information and the specific yeast strains under investigation.

#4 Elaborate a bit more on the deacidification abilities of Schizosaccharomyces yeasts. Provide a brief explanation of how this process occurs and its implications for the taste and aroma of cider. This would enhance the reader's understanding of why these yeast strains are of interest.

#5 Offer a bit more insight into why Sz. japonicus, despite having deacidification abilities, is not yet utilized in cider-making. Is there a specific reason for its underutilization, and does this study aim to fill that gap?

#6 Instead of placing the citation information (e.g., [1,2], [3]) within the text, consider integrating it more smoothly. For example, instead of stating "with a total revenue of $292.5 million in 2014 [3]," integrate the citation into the sentence structure for a more seamless flow.

#7 Explicitly mention the research gap or knowledge void that this study aims to address. What specific aspects of Schizosaccharomyces yeasts in cider-making are not well-understood, and how will this study contribute to filling that gap?

The results and discussion are well-constructed but some revisions have to be addressed:

#9 After presenting the results, explicitly discuss the significance of the findings. Explain how the observed differences in fermentation kinetics, pH, and compound production contribute to the overall quality and characteristics of the cider.

#10 When discussing fermentation kinetics, consider providing a brief explanation of why the duration of fermentation matters in cider production and how it influences the final product's characteristics.

#11 The substantial increase in pH with the Sz. japonicus strain is highlighted. Discuss the potential implications of such a pH increase on cider safety and quality. Additionally, consider relating this observation to previous studies or industry standards.

#12 When discussing the high pH, acknowledge the potential safety concerns mentioned and elaborate on how cider production processes typically manage or mitigate these concerns to ensure product safety.

#13 Connect the observed sugar consumption with ethanol production. Explain how the consumption of sugars influences the ethanol concentration and, consequently, the classification of the final product as apple cider.

#14 Provide more context on the significance of glycerol production by Sz. japonicus strains. Explain how glycerol impacts sensory properties and contributes to the overall taste and smoothness of cider.

#15 Further discuss the production of sorbitol and its potential impact on cider taste. Address any known sensory effects and consider linking this information to existing research or industry practices.

#16 When discussing volatile compounds, emphasize the aroma-active molecules. Explain how these compounds contribute to the sensory quality of cider and discuss the implications of their presence or absence.

#17 Discuss how terpenes contribute to the flavor profile of cider and whether the observed differences in terpene levels align with specific sensory attributes. Consider providing examples of how terpenes might be perceived by consumers.

The abstract is well written, but the quantitative informations are missing. Please rewrite your abstract

Round 2

Reviewer 1 Report

Comments and Suggestions for Authors

thanks for the revision

Reviewer 2 Report

Comments and Suggestions for Authors

Accept!